# Quantifying the distribution of feature values over data represented in arbitrary dimensional spaces

**Enrique R. Sebastian[‡]\*, Julio Esparza[‡]\*, Liset M. de la Prida**  **\***

Instituto Cajal, CSIC, Madrid, Spain

‡ These authors contributed equally to this work.
\* enrique.rodsebastian@gmail.com (ERS); esparzaj@cajal.csic.es (JE); lmprida@cajal.csic.es (LMP)

## Abstract

Identifying the structured distribution (or lack thereof) of a given feature over a point cloud is a general research question. In the neuroscience field, this problem arises while investigating representations over neural manifolds (e.g., spatial coding), in the analysis of neurophysiological signals (e.g., sensory coding) or in anatomical image segmentation. We introduce the Structure Index (SI) as a directed graph-based metric to quantify the distribution of feature values projected over data in arbitrary D-dimensional spaces (defined from neurons, time stamps, pixels, genes, etc). The SI is defined from the overlapping distribution of data points sharing similar feature values in a given neighborhood of the cloud. Using arbitrary data clouds, we show how the SI provides quantification of the degree and directionality of the local versus global organization of feature distribution. SI can be applied to both scalar and vectorial features permitting quantification of the relative contribution of related variables. When applied to experimental studies of head-direction cells, it is able to retrieve consistent feature structure from both the high- and low-dimensional representations, and to disclose the local and global structure of the angle and speed represented in different brain regions. Finally, we provide two general-purpose examples (sound and image categorization), to illustrate the potential application to arbitrary dimensional spaces. Our method provides versatile applications in the neuroscience and data science fields.

**Data Availability Statement:** All data used in this study is publicly available. For neural manifolds analysis, we use public data from the Collaborative Research in Computational Neuroscience Data

## Author summary

Many fields of science require analyzing data represented in high- and low-dimensional spaces in the form of point clouds. A common problem that emerges is how to quantify how features of interest are represented over these point clouds. In neuroscience, this problem arises while investigating neural representations, in the analysis of electrophysiological signals or in the segmentation of anatomical images. While many methods focus on characterizing the structure of the point cloud, there is a lack of approaches to quantify the distribution of feature values projected over the data points. Here, we introduce the Structure Index as a directed graph-based metric to quantify the distribution of feature values in a point cloud. Our method permits examination of the local and global

sharing platform as the th-1 head-direction dataset (http://crcns.org/data-sets/thalamus/th-1; doi:10.6080/K0G15XS1). For temporal series analysis, we used the public NSynth dataset at the TensorFlow Magenta project (https://magenta.tensorflow.org/datasets/nsynth), license CC BY 4.0. For image analysis, we used the Bird species dataset by Gerald Piosenka at the Kaggle platform (https://www.kaggle.com/datasets/gpiosenka/100-bird-species), CC0 1.0 Public domain. The code for simulations and figures is available at https://github.com/PridaLab/structure_index. An interactive notebook was developed to reproduce toy model data and analysis, and is available at: https://colab.research.google.com/github/PridaLab/structure_index/blob/main/demos/structure_index_demo.ipynb.

**Funding:** This work is supported by a grant from Fundación La Caixa (LCF/PR/HR21/52410030; DeepCode) to LMP. JE received the support of a PhD fellowship from "la Caixa" Foundation (ID 100010434; LCF/BQ/DR22/11950026). Access to supercomputer cluster Artemisa (project NeuroDIM) is co-funded by the European Union through the 2014-2020 FEDER Operative Programme of Comunitat Valenciana, project IDIFEDER/2018/048. The funders had no role in study design, data collection and analysis, decision to publish or preparation of the manuscript.

**Competing interests:** No competing interest to declare.

distribution of features, whether categorical/continuous or scalar/vectorial. Using case examples, we illustrate how the method can be applied to a wide range of data, from neural representations of the head-direction system, to sound and image categorization.

## Introduction

A point cloud is a prevalent data format found in many fields of science, which involves the definition of points in an arbitrarily high-dimensional space. Typically, each of these points is associated with additional values or features which require interpretation in the representation space. For instance, in neuroscience, neural activity can be pictured in a high-dimensional space, where each axis represents the activity of individual neurons [1]. Points in the cloud will follow the changes of population activity along time. In these so-called neural manifolds [2], one may project different features onto the point cloud, such as the animal's position or head-direction, as well as any other relevant behavioral variable [3]. In this context, understanding the local and global distribution of experimental features over the cloud can shed light onto neural representations such as those emerging during simple motor tasks [4] or those that support the internal navigational system [5,6].

Alternatively, temporal samples can be used directly to build an event space, where each axis corresponds to a timestamp. Examples include brain evoked potentials in response to natural sounds or speech [7] or internally generated brain waves such as sharp-wave ripples of the hippocampus [8]. Within this framework, each point in the cloud represents an event. By projecting the event characteristics, such as their frequency, onto the cloud, one can seek to separate different types of waveforms. Similarly, when segmenting images of single cells, each image can be represented as a point in a cloud residing in a space where each axis represents the intensity of a specific pixel and the features any label assigned to those images (e.g., the identity of cells, or any other anatomical information) [9]. Finally, in the realm of transcriptomic analysis, the point cloud originates from single cells in the gene space, and features could be the expression level of bona fide genetic markers or the biological state of the sample [10,11].

Having the ability to identify if and how a given feature is locally or globally structured along a point cloud can provide insights for an ample set of scientific applications. For instance, in deep neural networks trained for image recognition, object classes are represented globally while variations, such as angles, are encoded locally within each object class [12]. Similarly, behavioral features might map differently onto neural manifolds (e.g., speed, position, head-direction), providing new clues on how a particular brain region conjunctively represents information [13]. However, given the complexity and heterogeneity of datasets, a general-purpose tool to quantify these properties remains elusive. In some cases, one may be able to embed the original point cloud into a lower dimensional subspace to allow for visualization. Nevertheless, relying on visual inspection poses limitations when trying to compare across spaces or features, and becomes an obstacle when the analysis requires more than three dimensions. Moreover, whether the very same feature had structure in the original high-dimensional space typically remains unclear. Note that we use the term structure in a loose sense. That is, we say that a feature is structured across a point cloud if it follows any type of non-random distribution in the D-dimensional space.

Most methods focus on studying the structure of the point cloud itself, not on how an external feature is distributed across it. Available techniques aimed at describing feature distribution usually rely on clustering analysis [14]. Cluster cohesion is typically evaluated by

computing indices that compare intra- and inter-cluster distances, such as the Silhouette, Dunn and the Caliñski-Harabasz indices [15–17], while some methods seek to improve cluster analysis based on graph theory [18]. To inform on how an external feature is distributed across the point cloud using clustering one may evaluate whether the distribution of the feature values matches the defined clusters. However, when data points do not aggregate in groups, or the feature does not take discrete values, the resulting clusters are not directly interpretable. A method that does not rely on clustering is Mapper. It employs a function called *filter*, to represent the geometric properties of the point cloud in a graph [19]. While the *filter* function could potentially represent a feature to infer information about its distribution, Mapper in its current state does not offer this option and, importantly, it would only provide qualitative information.

Alternative methods resort to techniques that depend on linear correlation metrics, posing limitations for the analysis of more realistic convoluted distributions [20]. Other approaches based on decoders tacitly assume that if a given variable can be decoded from the data cloud, then it must follow some structure [21]. However, these strategies may be dependent on the model used, as well as on the intrinsic dimensionality of the data, being vulnerable to overfitting as sparsity increases with dimensionality. Crucially, all of these approaches provide poor insights into the local versus global structure of feature representations. It is therefore important to develop a method that (i) can be applied to non-linear distributions, (ii) generalizes to continuous features, and (iii) is applicable to arbitrarily high-dimensional spaces.

In this paper, we introduce the Structure Index (SI), a new metric specifically aimed at quantifying how a given feature is structured along an arbitrary point cloud. We first demonstrate the principles of our approach with simple model examples and illustrate how the method can be tuned to quantify the degree of the local/global organization of feature distribution, as well as its robustness along a broad range of data characteristics. Moreover, we show how the SI can be equally applied to vectorial features, in which more than one variable can be considered. Next, we apply the SI to neural data from experimental studies of head-direction cells, showing how it can retrieve representations of different features, which are quantified beyond visual inspection of the neural manifold. Finally, we provide two additional general-purpose examples (sound and image categorization), to illustrate the wide range of applications across fields.

## Results

### Definition of the Structure Index (SI)

The SI aims at quantifying the amount of structure present in the distribution of a given feature over a point cloud in an arbitrary D-dimensional space. For instance, feature values can be distributed in a 2D cloud along a gradient (Fig 1A), or randomly (Fig 1E). Identifying such structure without the need for visualization is a major problem in many applications, especially for high-dimensional spaces.

We start by considering a point cloud $P$ in an arbitrarily $D$-dimensional space, $P = \{p \in \mathbb{R}^D\}$, and a set of feature values f mapped onto $P$, $F = \{f_p | p \in P\}$. For now, we will consider f to be scalar ($f \in \mathbb{R}$) although, as we will explore later, we can extend this definition to higher dimensional features. The aim is to explore and quantify how the feature $F$ is distributed over the point cloud $P$. To do so, we first divide the point cloud $P$ into bin-groups ($B$) according to their feature value, such that $B_i = \{p \in P | t_i \leq f_p < t_{i+1}\}$, $P = \{B_1, B_2, \dots, B_n\}$, and $B_i \cap B_j = \emptyset | i \neq j; 1 \leq i, j \leq n$. That is, we create a finite covering $B = \{B_i\}_{i \in N}$ of the point cloud $P$ from these bin-groups.

Note that the exact way in which each bin limit is defined is open. One option is to define uniform bin limits as $t_i = (i - 1) * (max(F) - min(F))/n + min(F) | 1 \leq t \leq n + 1$, which is

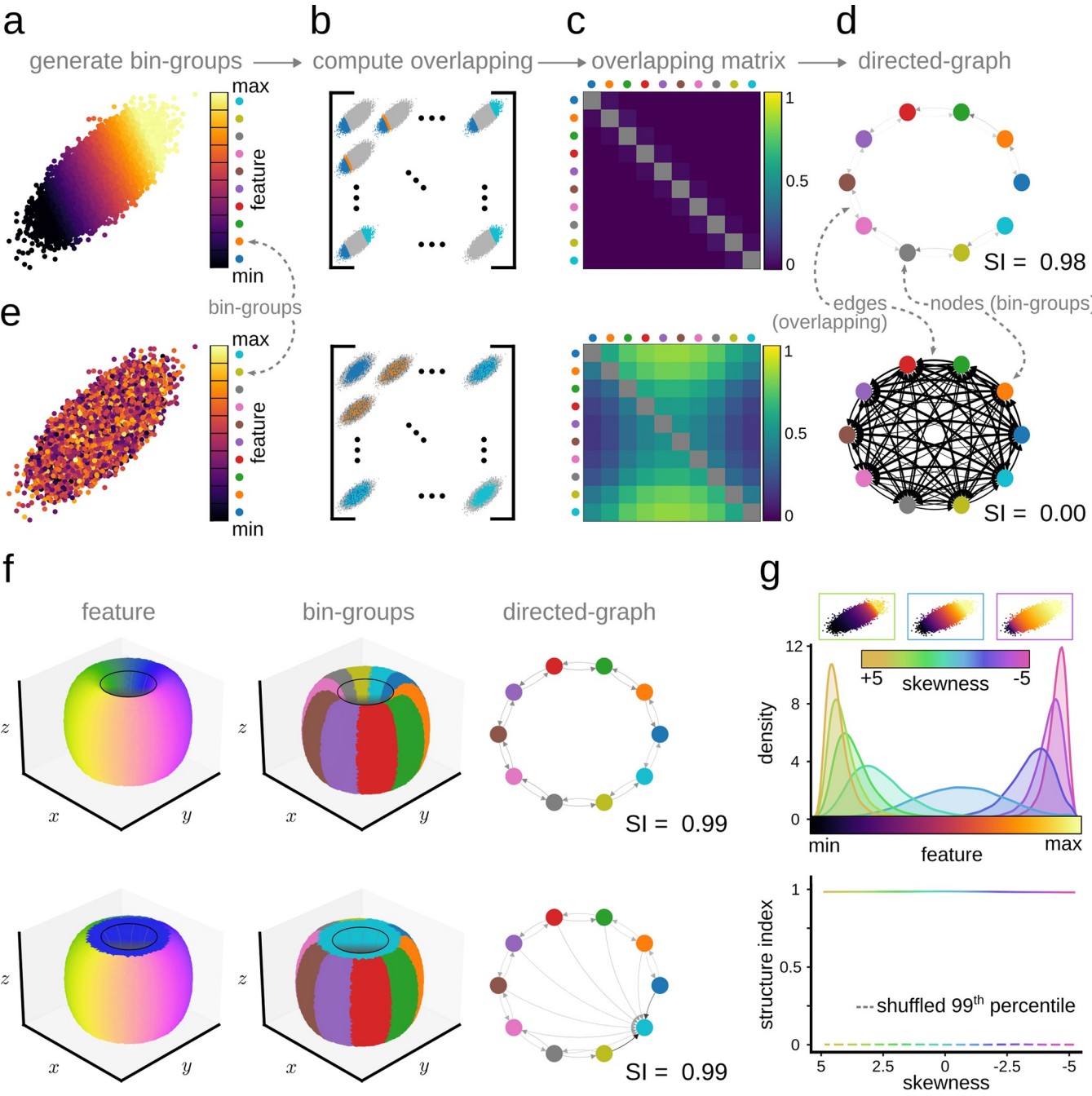

**Fig 1. Illustration of the concepts behind the definition of the Structure Index (SI). a** Feature gradient distribution in a 2D-ellipsoid data cloud. Each point in the data cloud is assigned to a group associated with a feature bin value (bin-group). **b, c,** Next, the overlapping matrix between bin-groups is computed. **d,** The overlapping matrix represents a connection graph between bin-groups, where structure (overlapping, clustering, etc..) can be quantified using the SI from 0 (random, equivalent to full overlapping) to 1 (maximal separation, equivalent to zero overlapping between bins). **e,** The case of a random distribution in a 2D data cloud. **f,** Different feature distribution over a torus yielding the same SI but different directed graphs. **g,** Lack of effect of the skewness of feature values on the SI. Continuous line represents the SI of the linear gradient for different levels of skewness. Dotted line represents the 99[th] percentile of a shuffled distribution.

the approach we will be taking throughout this work. Nevertheless, the method allows for defining the bin ranges in any custom way. Ideally, each bin-group should contain approximately an equal number of points (e.g., for features with a clear exponential distribution one may opt to use a logarithmic approach). Consideration of feature distribution (and hence of sample distribution within each bin) may become crucial for low-sample datasets and high number of neighbors.

Features can be either categorical (i.e., they may take nominal values associated to different categories) or continuous (i.e., they may take values within a scalar range). In case of a categorical feature, each bin-group may correspond to one of the possible discrete or nominal values the feature can take. Fig 1A shows the example of a feature distributed along a gradient in a 2D-ellipsoid data cloud as defined by the bin-groups shown on the right.

Next, we compute the overlap between each pair of bin-groups in terms of the $k$-nearest neighbors (Fig 1B). Given two bin-groups, $\mathcal{U}$ and $\mathcal{V}$, we define the overlap score from $\mathcal{U}$ to $\mathcal{V}$ ($OS_{\mathcal{U} \rightarrow \mathcal{V}}$) as the ratio of $k$-nearest neighbors of all the points of $\mathcal{U}$ that belong to $\mathcal{V}$ in the point cloud space. That is,

$$OS_{\mathcal{U} \rightarrow \mathcal{V}}(\text{k}) = \frac{1}{|\mathcal{U}| \cdot k} \sum_{u \in \mathcal{U}} |\{N_u^j(\mathcal{U} \cup \mathcal{V} - \{u\})|j = 1, \ldots, k\} \cap \mathcal{V}| \qquad (1)$$

where $N_u^j(\mathcal{U} \cup \mathcal{V} - \{u\})$ is the $j_{th}$ nearest neighbor of point $u$ in the set $\mathcal{U} \cup \mathcal{V} - \{u\}$.

Note that the definition of nearest neighbors is determined by the distance metric used (i.e., Euclidean distance, geodesic distance, etc.). Moreover, by defining an adequate distance metric, this method can be extended to complex point clouds and point clouds defined in non-Euclidean spaces.

Computing the overlap score for each pair of bin groups ($B_i$ and $B_j$) yields an adjacency matrix ($\mathcal{M}_{nxn}$) whose entry ($i,j$) equals $OS_{B_i \rightarrow B_j}$ (Fig 1C). $\mathcal{M}$ can be thought of as representing a weighted directed graph, where each node is a bin group, and the edges represent the overlap (or connection) between them (Fig 1D) [22]. Moreover, we do not allow any self-edges in the weighted directed graph i.e. we set $OS_{\mathcal{U} \rightarrow \mathcal{U}}(\text{k}) = 0$. Topologically, the set of bin groups ($B$) defines a finite covering of the point cloud. The nerve of this finite covering describes a one-dimensional weighted simplicial complex (i.e. the graph) whose vertex set is the indexing set of $B$ and whose directed edges are defined by the corresponding overlapping score. These mathematical notions may help to further develop a theoretical foundation of the SI metric in future work.

Finally, we define the Structure Index as 1 minus the mean weighted out-degree of the nodes after scaling it:

$$SI(\mathcal{M}) = 1 - \left( \frac{2}{n^2 - n} \sum_i^n \sum_j^n \mathcal{M}_{i,j} \right) \qquad (2)$$

Under this definition, for a uniform random distribution, the overlapping of any two nodes would be equal to 0.5 and therefore, the mean degree of the nodes of such a distribution would be 0.5. Thus, the Structure Index would take a value of 0 for a random distribution. In contrast, the mean degree of the nodes of a perfectly separated distribution would be 0 and thus, the SI would be 1. For small data sets and when using a small number of neighbors ($k$), the non-symmetry of $k$-nearest neighborhoods can yield slightly negative values which were set to 0. Therefore, the SI ranges between 0 (random feature distribution, fully connected graph) and 1 (maximally separated feature distribution, non-connected graph; Fig 1D).

By definition, the SI is agnostic to the type of structure (e.g., gradient, patchy, etc.) since bin groups do not need to follow any specific arrangement. Importantly, the SI directed graph

provides additional insights. Fig 1F shows the example of two different distributions with similar SI but different directed graphs. This aspect will be exploited below to illustrate how the SI graph can be used to infer additional data-driven information.

Finally, the skewness of feature values has little impact on SI, being robust for a wide range of statistical properties (Fig 1G).

Overall, our definition of SI and the corresponding graph makes this metric general enough to facilitate a range of applications. An interactive notebook was developed to reproduce toy model data and analysis: https://colab.research.google.com/github/PridaLab/structure_index/blob/main/demos/structure_index_demo.ipynb

Throughout the following sections, we will see the robustness of the SI metric under a wide range of datasets.

## Parameter dependence of SI on neighborhood size

To compute the overlap between each pair of bin-groups, the SI looks at the properties of the $k$-nearest neighbors of each point. For a low number of neighbors, the overlap is computed in the close vicinity of each point, thus being biased towards the local distribution. As the number of neighbors increases, the SI tends to better account for the global structure. This dependence of the SI on the number of neighbors can be exploited to infer information about the local versus global organization of data features.

Fig 2 shows two different feature distributions over the same data cloud that present with different patterning (local vs global). In the local pattern, feature values replicate over the different regions of the data cloud and so bin groups repeat across the regions (Fig 2A). In

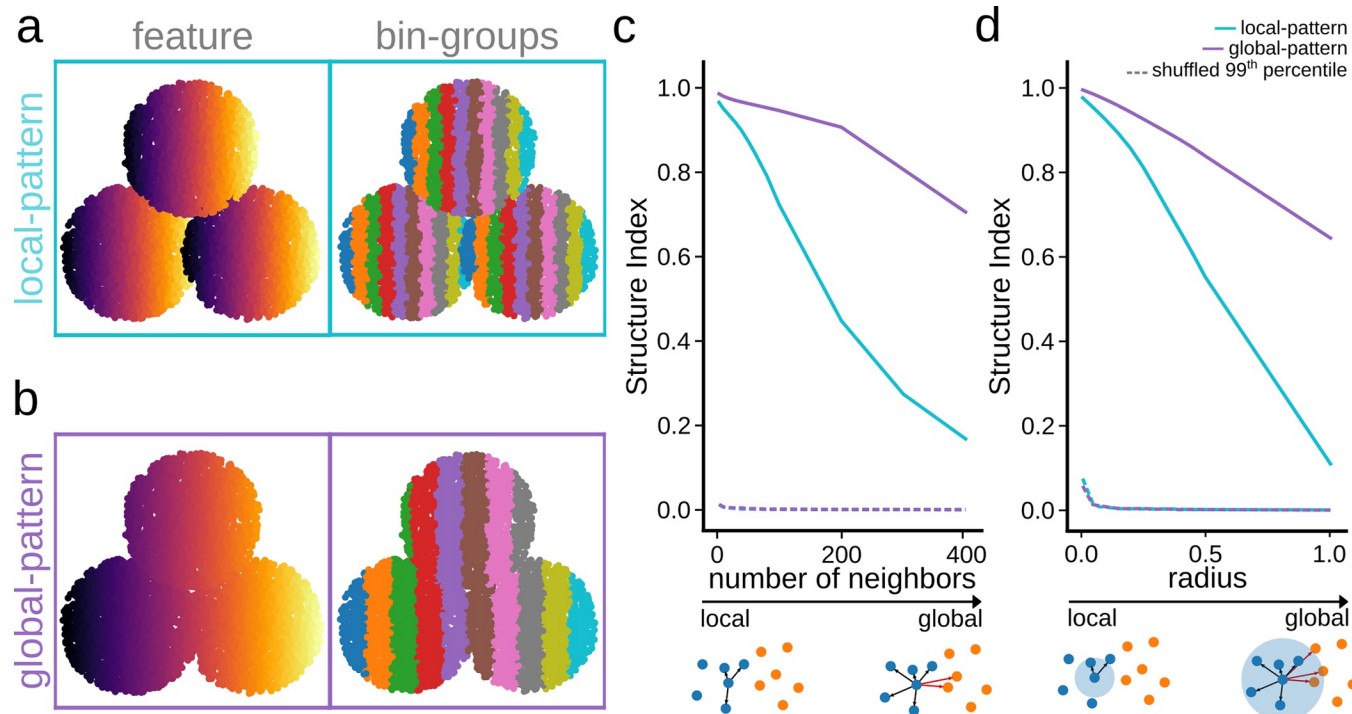

**Fig 2. Parametric dependence of SI on the number of neighbors. a**, A local pattern is simulated in 2D by projecting feature values differently along the data cloud (9000 points). Note local structure between bin-groups. **b**, The same 2D data cloud exhibiting a global distribution of feature values. **c**, Dependency of the SI values as a function of the number of neighbors can help identify the local versus the global feature distribution. Data was tested against a shuffled distribution of feature values (99th percentile). **d,** Same as in (**c**), but as a function of the radius.

contrast, in the global pattern, feature values follow a general trend (Fig 2B). By evaluating the evolution of the SI as a function of the number of neighbors, the trade-off between local and global structure can be quantified. For the local pattern, overlapping between bin-groups increases as the number of neighbors increase, and thus the SI sharply decreases. On the contrary, for the global pattern, the overlap is less sensitive to the number of neighbors, and therefore the SI decreases smoothly (Fig 2C). As expected, the SI of the shuffled distribution equals 0 independently of the number of neighbors. Thus, by tuning the number of neighbors, one can effectively change the sensitivity of SI to better detect local or global structures.

For data clouds with highly uneven density, the SI presents the option of setting a radius size ($r$) instead of the number of neighbors. Here, the neighbors of a point are set to be all points that fall within a given distance $r$. That is, Eq (1) becomes:

$$OS_{\mathcal{U} \rightarrow \mathcal{V}}(\mathrm{r}) = \frac{1}{|\mathcal{U}|} \sum_{u \in \mathcal{U}} \frac{1}{|\beta_u(r)|} |\beta_u(r) \cap \mathcal{V}| \tag{3}$$

$$where \; \beta_u(r) = \{x \in \mathcal{U} \cup \mathcal{V} - \{u\} : |u - x| \leq r\}$$

In such cases, the radius still helps to control for the trade-off between local and global structure, with smaller values making the SI more sensitive to local, and larger values increasing its sensitivity to global structure (Fig 2D).

## SI quantifies the distribution of scalar feature values

Before applying our method to the study of neural data, we tested it on toy model data to illustrate its performance and robustness for a wide range of point cloud characteristics. We generated 3 independent toy models, including: a) a 2D continuous linear gradient (40,000 points), b) a 3D solid ball (40,000 points) where the feature was distributed along the radius, and c) a 3D discontinuous cloud in the form of a lamp (32,000 points) whose feature varies along the three axes (Fig 3A).

To test for the stability of the method, we began by expanding these toy models into an increasing number of dimensions while adding white noise and then rotating the object in the extended space (Fig 3B). By doing so, we maintained the intrinsic dimension of the object but spread the information along all dimensions. The SI showed a consistent response while increasing dimensionality (Fig 3C). Importantly, the SI performed smoothly for a wide range of points in the cloud when examined in the original space (2 dimensions for the linear gradient, 3 dimensions for the other two objects) (Fig 3D).

In terms of the number of bin-groups used when computing the SI, there are two potential cases. For discrete or nominal features, the number of bin-groups is determined by the unique values the feature can take, so that there is a bin-group per discrete value. When dealing with continuous feature values, the number of bin-groups becomes a heuristic choice, which can be informed by statistical analysis. The number of bin-groups should be large enough so that the continuity of the feature values is fully captured, but small enough so that there is a reasonable number of points assigned to each bin-group. While the SI performs consistently for a range of bin-groups (Fig 3E), the topological characteristics of the data cloud may have different impacts that should be examined for each application.

Finally, we studied the sensitivity of the SI to different levels of noise in terms of the Signal to Noise Ratio (SNR), as defined in [23]. To this purpose, we introduced Gaussian noise across all existing dimensions (Fig 3F, left). While noise has an effect on structure, the SI was able to capture the trends even when introducing high levels of noise into the point clouds (Fig 3F, right). This renders the SI suitable for testing a wide variety of experimental data sets.

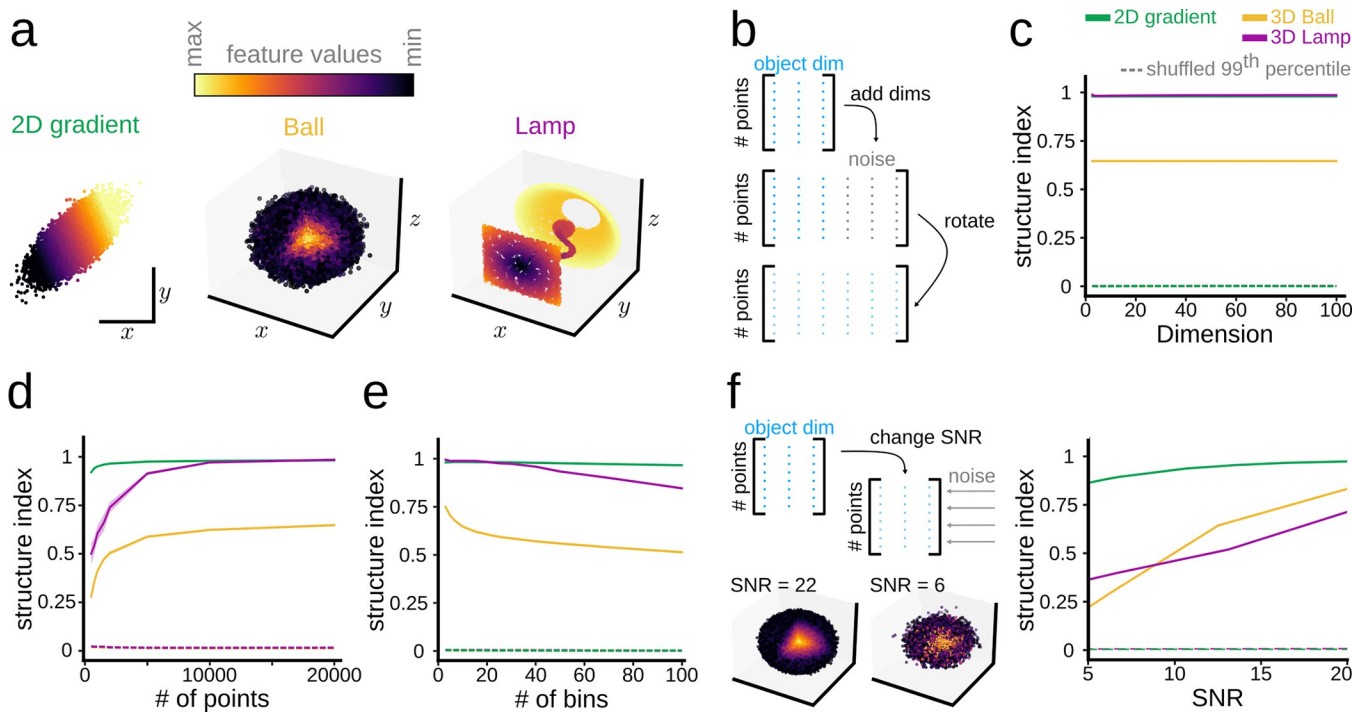

**Fig 3. Robustness of SI under a wide range of point cloud characteristics. a,** Three toy models used to evaluate performance of SI (40,000 points for the gradient and the ball; 32,000 points for the lamp). **b,** Objects in (**a**) were embedded into spaces of increasing dimensionality by adding noise and then rotating. **c,** Dependence of the SI on the embedded dimensionality for the three toy models. **d,** Effects of the number of points in the three data clouds. **e,** Effects of the number of bin-groups on the SI for the three toy models. **f,** Effect of different levels of SNR on SI.

## Evaluating the structure distribution of vector features

The definition of bin-groups used in the SI can be extended to vectorial features, which integrate values from several characteristics. For example, a vector feature (*A*, *B*) can be created from two scalar features, *A* and *B*, taking values along a continuous scale. In such a case, bin-groups can be defined by the upper and lower bound of both *A* and *B*. Thus, a point (*p*) in the cloud will fall within the bin-group *U* if and only if both entries of the associated vectorial feature fall within the common range.

To illustrate the case, we generated a point cloud sampled from a sphere of unitary radius using two angles *θ*, *φ*, with added Gaussian noise in 3D. Mathematically, the *x*-coordinate of a sphere is defined by the cosine of *θ*, while the *y*- and *z*-coordinates follow trigonometric relationships between both *θ*, *φ* (Fig 4A). Thus, a feature defined by *θ* and *φ* independently will distribute differently along the sphere from a vector feature defined by both angles (Fig 4B; top), and so will have different SI values. Hence, each of the three features will yield different bin-groups (Fig 4B; bottom). By definition, the structure of each angle separately should be lower than the vector angle (*θ*, *φ*). Moreover, given that the *x*-coordinate is completely defined by *θ*, we would expect more structure for *θ* than for *φ*. Consistently, the SI behaved as expected, with the lowest SI value obtained for *φ*, then for *θ*, and the highest value for both angles as a vector feature (*θ*, *φ*) (Fig 4C).

To evaluate the generalization of this behavior to vector features of any dimension, we generated point clouds sampled from D-dimensional spheres according to the equation shown in Fig 4D (left). For each point cloud in D-dimensional space, we computed the SI for both the *D*−1 angle used to generate the sphere and all angles together as a vector feature (Fig 4D, right). As predicted, the SI obtained when introducing all angles as a vector remained stable for all D-

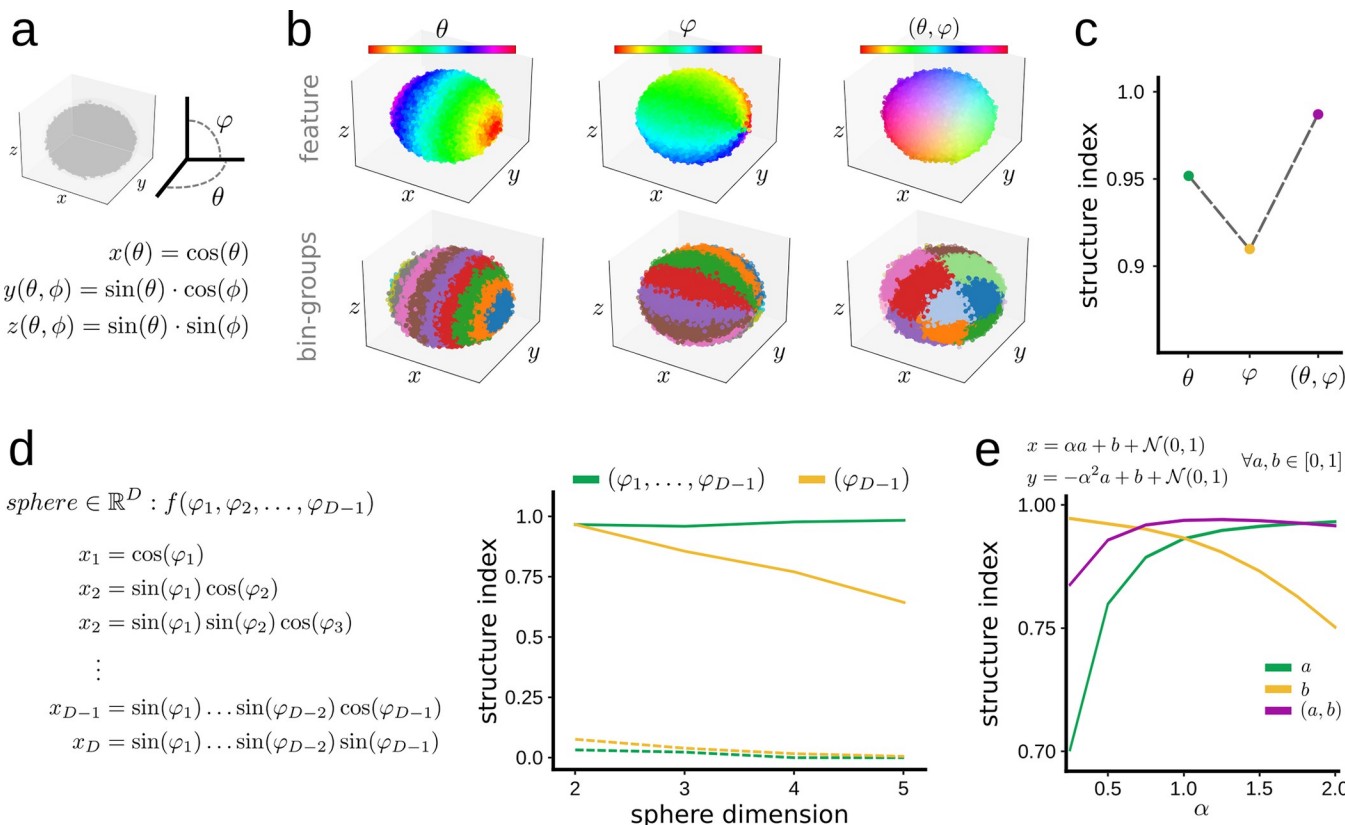

**Fig 4. Evaluating structure of vector features. a,** 3D sphere is defined by trigonometric equations depending on angles θ and φ. **b,** A synthetic point cloud was generated using equations shown in (**a**) (40,000 points). Feature values can be defined for each angle independently, θ or φ, and for both together in vector form (θ, φ). **c,** SI for each individual angle values and for the feature vector. **d,** A D-dimensional sphere is defined by trigonometric equations depending on D-1 angles ($8^{DxN}$ points, with N = 40,000 points to keep cloud density over N-dimensional spaces). The plot on the right shows the dependence of SI on the sphere dimension, computed for the D-1 angle alone, and for all angles in vector forms. Dashed lines indicate results from shuffled distribution values (99[th] percentile). **e,** Behavior of SI for a feature defined in 2D according to the equation shown at the top (20,000 points).

dimensional spheres. However, when only the D−1 angle was considered, the SI declined as the dimensionality of the sphere increased. This reflects the fact that as dimensionality increases, a lower percentage of coordinates depend on the D-1 angle, and thus the position of a given point is less dependent on it.

This property of the SI can be exploited to examine the interdependence between distinct interrelated features. For instance, we created a 2D cloud where the position of each point depends on two features: *a*, *b* (Fig 4E). While the impact of *b* on the position of the points was constant, the impact of *a* could be tuned by increasing or decreasing the parameter α. We proceeded by computing the SI of the scalar features *a*, *b*, and for the vector (*a*, *b*) using a range of α values (Fig 4E). The maximum SI(*b*) was obtained for α equal to zero (as the position of the points was completely defined by *b*) and decreased consistently as α increased. Accordingly, SI(*a*) increased with α as expected. Interestingly, SI(*a*, *b*) was lower than SI(*b*) for low α values (as the points are completely defined by *b*, the structure of (*a*, *b*) is lower than that of *b*). However, SI(*a*, *b*) rapidly increased with α reaching a plateau at 1 when both *a* and *b* equally contributed to the position of points.

These examples illustrate the capability of the SI to capture the structure of vector features, opening new avenues to study the relative impact and dependency between mathematically or experimentally related variables.

## Application to neural manifolds: the thalamic head-direction system

Having established the main readouts expected from the SI metric, we sought to apply it to the study of neural manifolds. To illustrate the effectiveness of the approach, we chose a public dataset of extracellular recordings from multi-site silicon probes in the anterodorsal thalamic nucleus (ADn) of freely moving mice [24]. This dataset has been recently used to demonstrate the intrinsic attractor manifold of the mammalian head-direction system [5], allowing us to directly test for the ability of SI to extract feature structure.

In this study, Chaudhuri et al. [5] showed that neural activity of N-simultaneously recorded ADn neurons of mice foraging in an open environment was constrained to a ring-shaped 3D manifold built with Isomap (Fig 5A and 5B; n = 6 mice). Therefore, structure was implicitly expected in the low dimensional representation. Whether similar structure was present in the original high-dimensional space however, was not tested. We therefore resorted to the SI to evaluate this point further.

Using the SI, we found structured distribution of the head angle in both the embedded and the original manifolds (Fig 5C; SI from 3 neighbors). When applied to the original N-dimensional neural space, the SI was slightly higher than in the low dimensional embedding for all mice examined (Fig 5C; grey box, paired sample t-test p = 0.011). Visualization of the individual weighted directed graphs from the high- and the low-dimensional representations confirmed similar feature organization (Fig 5D).

In their original work, the authors parametrized the manifold with splines of matching topology and used them to decode the represented latent variable (head-direction angle). We thus tested how the decoder performance (measured as the mean square error of predictions per mice) related to the head-direction information structured in the data. We found that the SI correlated with the decoder error (Spearman correlation -0.83, p = 0.042), following an exponential relationship ($R^2$ = 0.98; Fig 5E). That is, manifolds with lower decoding errors had higher head-direction structure as measured by SI.

Given the nature of the data, we wondered whether the head-direction representation can be retrieved during rapid eye movement (REM) as well as during slow-wave sleep (SWS) states [25]. To tackle this question, we included data from all states (awake, REM, and SWS) to compute the low-dimensional manifold. Indeed, when points of the manifold were color-coded according to the state, we noted some stratification (Fig 5F), which could be quantified with SI. The SI returned structure for both the animal state and the head angle, with higher values in the original than in the reduced space (Fig 5G; $F_{(2,1)}$ = 8.2, p = 0.007). Interestingly, structure was higher when using a vector feature consisting of the state and head angle together, indicating that there may be some interdependency between them (Fig 5G).

Finally, we computed the SI of the head angle for each state separately (Fig 5H). Consistent with data above, SI was maximal in awake conditions (ANOVA effects for state ($F_{(2,1)}$ = 83.5, p<0.0001), space ($F_{(2,1)}$ = 25.7, p<0.0001) and interaction). Moreover, although REM and SWS yielded low angle structure in the manifold, the SI values were significantly higher in the original space, potentially indicating that information was lost while reducing dimensionality. Thus, the SI allows for quantitative comparison of neural representations during different behavioral states and importantly, permits the evaluation of this neural activity in the original space.

## Application to neural manifolds from different brain regions

The head-direction system encompasses multiple interconnected brain networks, including the brainstem, ADn, post-subiculum (PoS), and the entorhinal cortex [26]. Interestingly, previous work using the same dataset shown in Fig 5 but including both ADn and PoS neurons,

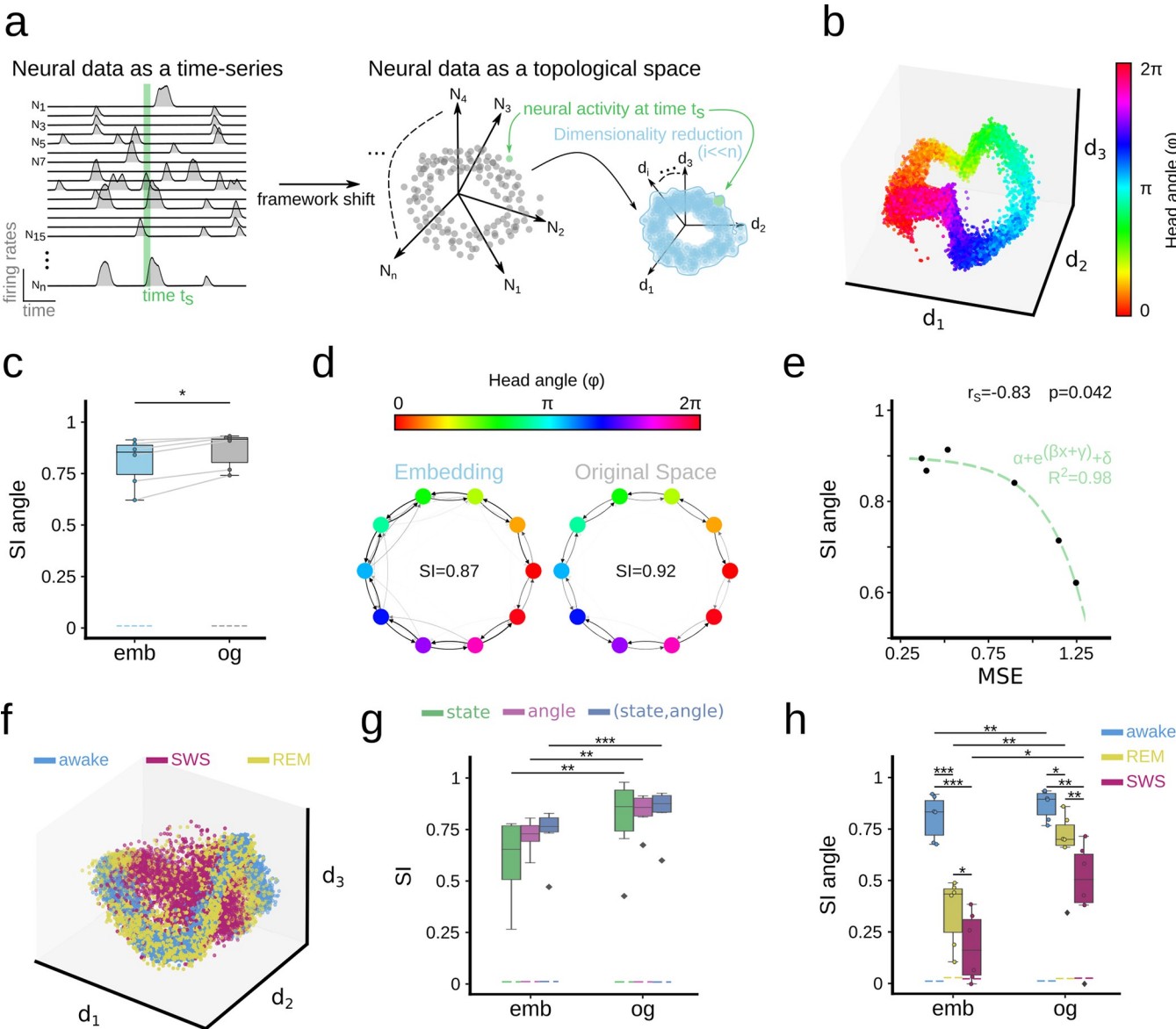

**Fig 5. Using SI to evaluate neural representations. a,** In the neural manifold framework, firing rates from N-neurons at a given time ($t_s$) are represented in an N-dimensional Euclidean space. Activity is constrained in a subspace which can be retrieved using dimensionality reduction approaches (d-dimension). **b,** Neural manifold computed from the head-direction system by Chaudhuri et al.[5], with the head-direction angle projected over the data cloud. **c,** SI of the head-direction angle in the original (og) and embedded representations (emb). Dashed lines indicate results from shuffled distribution values (99th percentile). **d,** Example of the weighted directed graphs from the same mouse in the original and in the low-dimensional embedding. Note similar organization. **e,** Relationship between the SI and the mean square error (MSE) of the decoder trained by Chaudhuri et al.[5] in the reduced space. Fitting curve parameters: $\alpha = -0.12$; $\beta = 4.47$; $\gamma = -4.7$ and $\Delta = 0.9$; tested significant at $p < 0.05$. **f,** Head-direction data plotted over the 3D embedding for awake, REM, and SWS states separately. **g,** SI for states and angles separately, and for both features together as expressed in vector form. Results are shown for both the original (og) and the reduced space (emb). ANOVA effects for space ($F_{(2,1)} = 8.2$, $p = 0.007$) but not for feature nor interaction. Post-hoc tests: **, $p < 0.001$; ***, $p < 0.0001$. **h,** SI of the head direction angle for each state separately both in the original (og) and the reduced space (emb). ANOVA effects for state ($F_{(2,1)} = 83.5$, $p < 0.0001$), space ($F_{(2,1)} = 25.7$, $p < 0.0001$) and interaction. Post-hoc tests: **, $p < 0.001$; ***, $p < 0.0001$.

found some residual low-dimensional structure that correlated with speed of movement [21]. Thus, we next decided to apply the SI to evaluate the structure of different features over neuronal manifolds built from different brain regions.

We computed the SI in the original space of ADn and PoS neurons separately and together (ADn+PoS), and found different structure for the head angle and mouse velocity (two-way

ANOVA for features: F(2,2) = 80.8, p<0.0001; region: F(2,2) = 5.7, p = 0.0125; and interaction F(2,2) = 3.5, p = 0.0286). The ADn neural manifold presented a larger SI for the head angle than the PoS manifold, which in contrast displayed more structure for velocity than ADn neurons (Fig 6A). In the joint ADn+PoS neural space, the SI for angle and velocity reached similar levels to those for individual regions suggesting that information is fully recovered in the joint manifold. Consistently, the overall structure was larger when considering the vectorial feature (angle, velocity) (Fig 6A). These results cannot be explained by an unbalanced contribution of cells from the different brain regions (see Methods).

Next, we computed whether feature structure was preserved in the manifolds built at increasing dimensional spaces (Fig 6B; from 1 to 10 dimensions). The SI of head angle sharply plateaued at 2 dimensions for ADn and ADn+PoS (consistent with a 2D ring organization), whereas PoS followed a smoother trend up to 4 dimensions (Fig 6B; left). In contrast, SI of mouse velocity plateaued at 3 dimensions for PoS, whereas ADn presented a much lower structure that increased up to 6 dimensions, similar to ADn+PoS (Fig 6B; center). Structure of

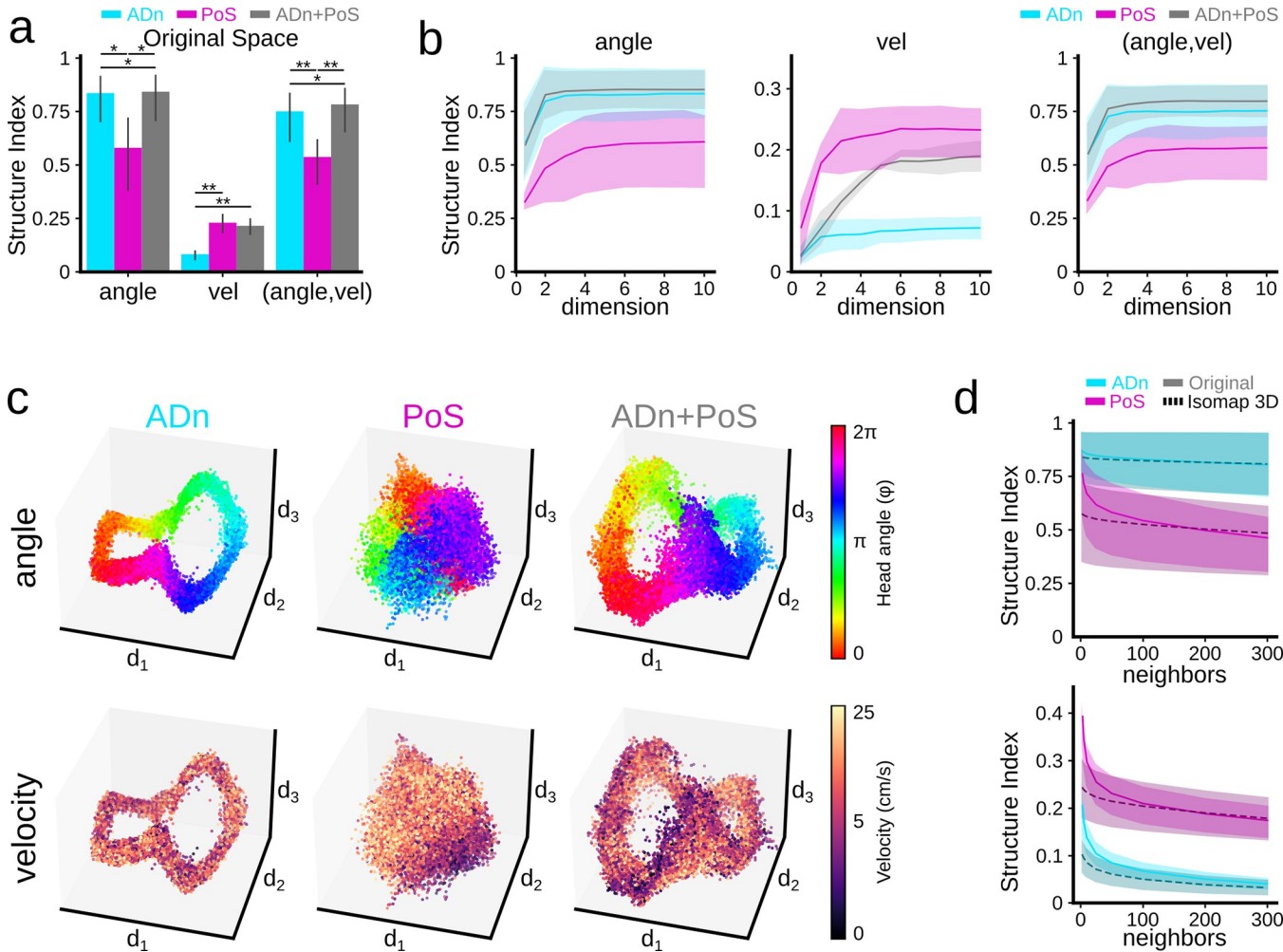

**Fig 6. Using SI to evaluate neural representations across brain regions. a,** SI in the original space of the head direction angle, velocity, and (angle, velocity) for ADn, PoS, and ADn+PoS. Two-way ANOVA for features: F(2,2) = 80.8, p<0.0001; region: F(2,2) = 5.7, p = 0.0125; and interaction F(2,2) = 3.5, p = 0.0286. Poshoc tests *, p<0.05; **, p<0.001 **b,** Relationship between SI and the number of Isomap dimensions used to embed the neural activity of the different brain regions. **c,** Example of Isomap 3D embeddings from the different brain regions on the same mouse. Color coded by head angle (top row) and velocity (bottom row). **d**, SI for angle (top) and velocity (bottom) as a function of the number of neighbors (local vs global), for ADn and PoS.

the vectorial feature (angle, velocity) confirmed that information could be properly retrieved for up to 3 dimensions in all cases (Fig 6B, right). This suggests that SI can be exploited to evaluate the quality of the dimensionality reduction of neural manifolds.

These trends can be visualized in the 3D embedding from all manifolds colored by the head angle (top) and velocity (bottom) (Fig 6C). Consistent with previous analysis, the ADn manifold exhibited a ring shape with a clear distribution for the angle but not for speed (Fig 6C, left). Instead, the PoS embedding showed a compact tridimensional shape, where the head angle is encoded in a circular manner and mouse velocity follows a gradient distribution (Fig 6C, middle). Finally, the ring-shaped ADn+PoS embedding encoded both the angle and velocity albeit in different directions (Fig 6C, right), consistent with the larger structure found for the vectorial feature.

To further illustrate how the SI can help evaluate the local/global nature of neural representations in the ADn and PoS regions, we computed SI in both the original space (continuous line) and in the 3D embedding (dashed line) for an increasing number of neighbors (Fig 6D). For ADn, SI of the angle was almost constant, consistent with a smooth global distribution (Fig 6D, top). Instead, SI of mouse velocity decreased sharply with increasing number of neighbors, suggesting a local organization (Fig 6D, bottom). Interestingly, PoS showed a more local structure for both angle and velocity, especially in the original space. These results confirm that the SI metric can be used to inform quantitatively about the structure of different features distributed over neuronal manifolds from different brain regions.

## Application to arbitrary D-dimensional spaces: temporal samples and images

Finally, to illustrate applicability of the SI metric for analysis of other types of data and across fields we applied the SI to two additional general-purpose examples: sound (temporal data) and image categorization (pixel data).

For temporal data, we resorted to musical notes given similarities with electro-encephalographic event waveforms [27]. Using this dataset allowed us to focus on evaluating SI performance directly, given interpretability of the music. Data samples consisted of 4 second snippets of notes of different pitch and velocity, down-sampled to 4800 time stamps. They were produced by different instruments (including the human voice) using acoustic, electronic or synthetic sources. Instruments were annotated as belonging to different families. Notes were represented in the 4800-dimensional space (S1A Fig), and the SI was calculated both locally (using 3 neighbors) and globally (60 neighbors). In general, data showed a higher local than global structure (S1B Fig; $F_{(3,1)} = 4.0$, p<0.0001). We found that the pitch provided maximal structure, followed by the source and the instrument family, as confirmed by the directed graph (S1C Fig; ANOVA effects for features $F_{(3,1)} = 12.0$, p<0.0001). Reducing data to 3D allowed for visualization of these trends (S1D Fig), and provided similar SI trends as for the original space (S1E Fig).

This example illustrates the power of the SI graph directionality, which enables investigation of the links between notes. Specifically, asymmetric connections similarities can be identified among different instrumental families (S1C Fig). For instance, vocal samples (notes produced by human voice) seem to overlap asymmetrically with multiple other families. This might be due to the great variability of tones produced by the human voice. This versatility may cause a great scattering of the location of the vocal data points depending on those nuances, such that sometimes they more resemble a guitar (string based) or a flute (wind based). Indeed, by leveraging the directed graph, one may interpret the asymmetry in these overlaps as an indication that it is the vocal points that are infiltrating into the other families, rather than mixing.

To explore SI application in high-dimensional spaces, we chose images of bird species, typically exploited in fine-grained recognition problems. The data consisted of RGB channels (56x56x3 pixels) so that each image was represented as a point in a 9408-dimensional space (S2A Fig). Birds were classified as belonging to different species, geographic continent, scientific order and family. SI was maximal for bird species, followed by family and order (S2B Fig). We noted that continents provided the lower structure, potentially reflecting migratory habits and/or species diversification. Visualization of images showing maximal and minimal overlapping values confirmed that the SI successfully captured the underlying structure of the data (S2C and S2D Fig).

These examples illustrate how the SI method successfully operates in arbitrary D-dimensional spaces, allowing for its widely ranging application to various fields of science.

## Discussion

With the development of a directed graph-based topological metric (SI), we have enabled accurate quantification of the structure of features distributed over point clouds. The approach is not constrained by the dimensionality of the space and is robust to a wide range of data and feature characteristics. Importantly, the SI metric not only quantifies the "amount" of structure of scalar feature values, but it can also provide insights into the topological distribution of the feature by looking at the overlapping directed graph. Moreover, the resulting weighted-directed graph opens possibilities for further graph-based analysis.

A common issue in current dimensionality reduction methods is being able to capture the global structure without deforming local relationships. Indeed, most dimensionality reduction methods have a parameter to control for that tradeoff (e.g. the number of neighbors). Here, we demonstrate that the SI can be tuned to better detect local vs global structure by changing the number of neighbors (or equivalently the radius) used to compute overlap between bingroups. Thus, the SI can be used not only to quantify the structure in the original space, but also to evaluate the quality of the dimensionality reduction by looking at how much structure has been preserved both locally and globally by different dimensionality reduction methods.

The SI metric is not in direct competition with the many methods that use cluster analysis [14–17]. Rather, it generalizes to a class of distributions (i.e. continuous distributions) where clusters typically fail to apply. To handle this, some researchers discretize continuous variables or use some discrete states for categorization [28], but in many cases there is no clear separation and the categories are shown to strongly intermix over the continuous point cloud [29,30]. This complicates interpretation, especially for experimental features and data-driven approaches. While other methods like Mapper or topological decoders can be applied to continuous point clouds [19,21], they both differ from our approach. For topological decoding, the goal is to map the distribution of an observable (i.e. a feature) over the point cloud so that it can be reconstructed. However, this provides no quantitative information on how the feature is actually distributed, as the SI does. On the other hand, Mapper focuses on how the point cloud is structured in relation to the *filter*, while the SI evaluates how a given feature is structured along the point cloud. If one were to define Mapper's *filter* as the external feature, the resulting graph would potentially convey similar information albeit with some important differences. First, in contrast to SI, the graph produced by Mapper is undirected, giving poor information about the feature distribution in topologically similar point clouds (e.g., Fig 1F). Secondly, while Mapper only offers qualitative information, the SI metric provides an index that quantifies the local and global structure—a crucial aspect when comparing datasets.

As demonstrated above, the SI can be extended to vector features, expanding the range of applications. Note that a vector feature can be created by grouping multiple scalar/categorical

features, or by integrating several related variables. By doing so, the SI allows for the study of how different features interact with each other, allowing for a deeper understanding of how data structure is determined. This may ease data-driven discoveries of latent interaction between experimental features, which cannot be established a priori.

In this context, we have applied the SI to study the representational capability of the head-directional system [24]. By using data from a previous study that demonstrated low dimensional representations of the head-direction angle in ADn cells [5], we have shown that the SI captures structure both in the lower dimensional manifold and in the original space. More-over, by applying this metric to awake, REM, and SWS states, we showed that the head-direction representation is preserved during sleep, providing additional interpretations. This observation is consistent with recent data supporting mental recreation of head-direction angles during REM sleep [25] and provides backing for the use of the SI metric in helping to infer mechanisms underlying neural manifolds. Importantly, the SI is not limited to manifolds built in Euclidean spaces, as one can define the $k$-closest neighbors in terms of different distance metrics, such as geodesic or hyperbolic distance among others [31,32].

We also exploited the SI metric to examine the structure of angle and velocity over neural manifolds built from different regions of the head-direction system. We found striking differences in the local versus global structure of feature values encoded independently from ADn and PoS, confirming their complementary roles [26]. Moreover, by using the SI we were able to evaluate the contribution of mixed activity from the different regions in a common ADn +PoS manifold. Consistent with topological methods, the SI was able to quantify the low-dimensional structure correlated with speed in the head-direction system [21]. This further illustrates the utility of the SI to extract biologically meaningful information in support of data-driven discoveries.

Additionally, the SI method can be applied to the study of temporal data expressed in high-dimensional spaces. By representing temporal events in the axes built from individual time stamps, electrophysiological signals can be analyzed with state space methods [33–35]. Applying the SI to these space representations may thus allow for new strategies for the analysis of the spectro-temporal organization of brain oscillations [7,8,30]. Similarly, the SI permits image quantification and categorization in the service for fine-grained image recognition problems applicable to several research fields.

As topological and high-dimensional analysis become the norm in the neuroscience field, we expect that the SI will be a powerful tool to shed light onto a wide range of questions. Here we have provided several examples, from high-dimensional geometrical analysis to sound and image categorization, expanding the applicability of the tool across fields.

## Materials and methods

### Datasets

In this study, we used different datasets to evaluate SI performance. For the parameter study, we created objects (2D-ellipsoids, balls and spheres) using the corresponding mathematical equations. To generate the ball and sphere point clouds, points were generated by uniformly sampling the angles (and radius, if applicable), then computing the corresponding Euclidean locations (from those angles) and adding Gaussian noise to the resulting points. For the object lamp, we used the model from the ModelNet40 dataset, which is publicly available at https://github.com/antao97/PointCloudDatasets. Different feature value distributions were created over these objects and used to evaluate SI performance. By default, all objects were created with 40,000 data points, except where otherwise reported. To study how SI behaved under

noise (Fig 3F), we added different levels of Gaussian noise to the location of the points by using the 'multivariate_normal' function from the scipy.stats Python library.

To study neural manifold representations, a publicly available head-direction dataset was used (http://crcns.org/data-sets/thalamus/th-1; doi:10.6080/K0G15XS1) [24]. We chose this dataset because the neural manifold organization of head angles was recently validated [5], excluding any confounding in the ability of the SI to extract structure. Moreover, as we will show in the Results section, using these data allowed us to illustrate the capacity of the SI to quantify structure in the original space, which was not tested in the aforementioned reference due to lack of computationally efficient methods available. We used all data available to build the 3D neural manifold as reported in [5] using Isomap [36]. We also built the representations in the original space using single cell data (one axis per cell), yielding different high-dimensional spaces per mouse using data from ADn (n = 6; mouse 1: 37 cells; mouse 2: 29 cells; mouse 3: 11 cells; mouse 4: 10 cells; mouse 5: 10 cells; mouse 6: 22 cells). Information about brain states was used to separate the ADn neural manifold from awake and sleeping periods, with sleep classified as SWS or REM sleep. We also built manifolds from animals that had neurons simultaneously recorded from both ADn (n = 3; mouse 4: 10 cells; mouse 5: 10 cells; mouse 6: 22 cells), and PoS (n = 3; mouse 4: 13 cells; mouse 5: 15 cells; mouse 6: 38 cells). Finally, to evaluate the contribution of different regions, we examined the joint manifold of those animals using data from both regions together, ADn+PoS (n = 3; mouse 4: 23 cells; mouse 5: 25 cells; mouse 6: 60 cells). Note that except for mouse 6, the contribution of ADn and PoS neurons to the joint manifold was relatively balanced.

To evaluate the application of SI to temporal data, we opted for musical notes given their similarities with electroencephalographic waveforms [27]. An additional advantage is that musical notes are directly interpretable, allowing us to focus on evaluating SI performance. We chose the NSynth dataset [37], which contains over 300000 musical notes produced by around 1000 different acoustic, electronic or synthetic instruments, including the human voice. This dataset is available on the TensorFlow Magenta project at https://magenta.tensorflow.org/datasets/nsynth. Different features (source, instrument family, pitch and velocity) characterize each musical note. Each note consists of 4 second monophonic 16 kHz audio snippets at five different velocities. For analysis, we downsampled the original snippets to 1.2 kHz resulting in 4800 time-stamps, which were used to build the high-dimensional space (one point per note). To comply with the Nyquist–Shannon sampling theorem, audio snippets with an associated pitch higher than 73 were discarded. Binary features were not included in the analysis. For statistical testing, the dataset was divided into 5 independent batches, which were analyzed both in the original and the 3D-reduced space using Uniform Manifold Approximation and Projection (UMAP), which is known to provide better results for high-dimensional data [38].

Finally, to provide examples of image analysis using SI we resorted to the bird species problem, given its application in fine-grained image recognition. Similar to above, this allowed us to focus on evaluating the performance of SI directly. To this purpose, we used the 100-bird species dataset created by Gerald Piosenka, which is hosted on the Kaggle platform (https://www.kaggle.com/datasets/gpiosenka/100-bird-species; date of download July 28th, 2022). The dataset consists of more than 70000 RGB images of 450 bird species. Images are 224 x 224 pixels x 3 color (jpg format) annotated by species name. For analysis, we downsampled images to 56x56 pixels x 3 colors, resulting in a 9408-dimensional space, where each axis is the value of a pixel (one point per image). To expand the number of features associated to each image, we performed an automated data scraping from Wikipedia, so that for each bird species we also extracted information about geographical distribution (continents), as well as the scientific order and family, using the Python library 'Wikipedia' (https://pypi.org/project/wikipedia/).

## Computational resources

All simulations and analysis were performed in Python 3.8.13 using personal computer workstations (Intel Xeon CPU E5-2620 v4 @ 2.10GHz processor with 16 cores, 64GB RAM memory, GeForce GTX 1080 Ti GPU with 11GB memory and 0.355 TFlops for double precision). Whenever required, the supercomputer cluster Artemisa (https://artemisa.ific.uv.es/web/content/nvidia-tesla-volta-v100-sxm2) was used to accelerate calculations (NeuroDIM Project).

## Statistical analysis

Statistical analysis of different instances of each dataset was performed using one- or two-way ANOVAs followed by Student t-tests or equivalent.

## Supporting information

**S1 Fig. SI applied to temporal data. a,** Data consists of musical notes from different instruments, which can be represented in a 4800-dimensional Euclidean space, where each axis is one timestamp. Notes of similar pitch are expected to lie closer in the high-dimensional space, with family instruments providing some additional structure. **b,** SI of the different features of the musical notes (source, instrument family, pitch, and velocity) both in a local (3 neighbors) and global (60 neighbors) region in the original space. Dashed lines represent 99th shuffled percentile. Global structure was in general lower as compared with local structure. Note higher structure for the pitch versus the source and instrument family. **c,** Directed weighted graph returned by the overlapping of instrument family. Note nodes of similar instruments located closer in the graph. Note that direction and width of connecting edges give information on instrument similarity. **d,** Projection of the different categorical and continuous features in a 3D embedding created with UMAP from the original 4800-dimensional space. Note larger structure for pitch, source and family. **e,** SI of the different features of the musical notes in the 3D embedding both in a local (3 neighbors) and global (60 neighbors) vicinity. Dashed lines represent 99th shuffled percentile.
(TIF)

**S2 Fig. SI applied to image classification. a,** Data consists of annotated RGB images (56x56x3) of multiple bird species. Each image can be represented as a point in a 9408-dimensional Euclidean space where each axis is the value of a pixel. **b,** SI of the different features from each image, including the species, continent, scientific order and family. Dashed lines represent 99th shuffled percentile. **c,** Examples of species of birds showing maximal (Fairy Bluebird) and minimal overlap (Northern Fulmar) with Cape Glossy Starling. **d,** Examples of species showing maximal (Hornbill) and minimal overlap (Golden Pipit) with King Vulture. Bird images are reproduced from the Bird-species dataset by Gerald Piosenka at the Kaggle platform (https://www.kaggle.com/datasets/gpiosenka/100-bird-species), license CC0 1.0 Public domain.
(TIF)

## Acknowledgments

We thank Erik Thordsen for useful comments and suggestions.

## Author Contributions

**Conceptualization:** Enrique R. Sebastian, Julio Esparza, Liset M. de la Prida.

**Formal analysis:** Enrique R. Sebastian, Julio Esparza.

**Funding acquisition:** Liset M. de la Prida.

**Investigation:** Enrique R. Sebastian, Julio Esparza, Liset M. de la Prida.

**Methodology:** Enrique R. Sebastian, Julio Esparza.

**Project administration:** Liset M. de la Prida.

**Supervision:** Liset M. de la Prida.

**Writing – original draft:** Julio Esparza, Liset M. de la Prida.

**Writing – review & editing:** Enrique R. Sebastian, Liset M. de la Prida.

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
