## [Decision Letter · Decision Letter 0]

26 Jul 2023

Dear Dr Menendez de la Prida,

Thank you very much for submitting your manuscript "Quantifying the distribution of feature values over data represented in arbitrary dimensional spaces" for consideration at PLOS Computational Biology.

As with all papers reviewed by the journal, your manuscript was reviewed by members of the editorial board and by several independent reviewers. In light of the reviews (below this email), we would like to invite the resubmission of a significantly-revised version that takes into account the reviewers' comments.

The reviewers feel the manuscript is interesting, but have some major concerns that must be addressed. In your revision, please pay special attention to the comments about comparing your method to Mapper (Rev 1) and other methods for structure quantification (Rev 2), as well as comments about the lack of a theoretical foundation.

We cannot make any decision about publication until we have seen the revised manuscript and your response to the reviewers' comments. Your revised manuscript is also likely to be sent to reviewers for further evaluation.

Sincerely,

Carina Curto

Guest Editor

PLOS Computational Biology

Thomas Serre

Section Editor

PLOS Computational Biology

The reviewers feel the manuscript is interesting, but have some major concerns that must be addressed. In your revision, please pay special attention to the comments about comparing your method to Mapper (Rev 1) and other methods for structure quantification (Rev 2), as well as comments about the lack of a theoretical foundation.

Reviewer's Responses to Questions

**Comments to the Authors:**

Reviewer #1: One of the very general forms of data encountered in many fields is that of a point cloud with "some data" or "some feature" attached to it. Understanding how this feature is structured, and, more importantly, how its structure relates to the structure of the underlying point cloud, is therefore an important goal of data analysis, with broad implications. Motivated by this, the authors construct what they call the "Structure Index (SI)", and show how it can help answer parts of the feature structure problem in both synthetic and real datasets.

Their experiments are quite convincing, and although there is no theoretical foundation, it does from these experiments seem plausible that the SI is a useful quantity. However, there is one major issue I see: The SI seems *extremely* related to (parts of) the well-established construction known as Mapper [Singh, G., Mémoli, F., & Carlsson, G. E. (2007). Topological methods for the analysis of high dimensional data sets and 3d object recognition. PBG Eurographics, 2, 091-100.]. I am not certain, but it feels to me that the SI (and perhaps generalizations of it) can be derived from some variation of Mapper. Even if I am mistaken, the (superficial?) similarities of the two methods, and the degree to which Mapper is well-established, demand that the paper contain an explicit comparison.

Minor issues follow:

* It should be made clear exactly what random process created the point clouds. This is particiularly true for those (e.g. spheres) with a non-identity parameterization.

* In the investigations of noise, it's not clear whether noise was added to just the features, or to the point positions, or to both. Either is probably fine, but should be made clear.

* The subfigure labels in figure 4 seem confused?

* Did you experiment also with non-uniform bins? Perhaps explain why uniform bins are chosen.

Reviewer #2: In this manuscript, the authors present a metric, SI, designed to quantify how feature values are distributed over point clouds, such as those from large neural recordings. Given the speed with which increasingly large neural data sets are appearing, there is a strong need for such metrics/approaches to understand and explore the data. The authors develop their approach through applications to simulated and real data with varying structural and noise properties. While I did not find the theoretical arguments for the metric to be particularly compelling, it does seem to capture interesting properties of the data in a simple and potentially useful way and would suggest further investigation.

Suggestions:

1) While the method is presented in an intuitive way, there is a lack of theoretical foundation (proofs, derivations, etc.), which makes it difficult for the reader to understand, e.g. in what sense “SI quantifies the topological distribution of scalar feature values.” I would expect such a claim to include a clear argument and, in my opinion, such an argument for the metric would greatly improve the manuscript. Alternatively, the authors could make it clear that such an argument is still lacking, suggest this as an avenue for future work, and tone down the language.

2) The authors use some terms in a non-standard way and without defining them sufficiently for the reader, e.g. “topologically organized”, “manifold”, and “representational capacity”, all appear in a sentence in the first paragraph. I would suggest simplifying the terminology throughout and including more definitions.

3) I would recommend giving a simple example early in the introduction, maybe something with head direction as the feature, and explaining what the SI could capture about how HD is represented across the population. It currently takes the reader a while to understand what is being referred to as a feature, etc, and what the goal is of the work. Also, such a running example through the text might help walk the reader through some of the other concepts. For example, in the local versus global, I would imagine that while the standard HD tuning would appear global, one could construct HD simulations where the local/global organization is different despite having a valid HD code (e.g. maybe neurons could have tuning curves where the firing rate could move abruptly over angles such that neighboring angular bins would have very different population vectors, or maybe HD could be represented by multiple independent populations that become active/inactive independently… thus one could vary how HD is distributed over the population while maintaining its decodability). This might also help illustrate that the measure is different than decodability (e.g. fig 5(e)). Then again, in the discussion of the vector features, perhaps the example could be extended to velocity (HD*speed), which could then illustrate how the measure appears when speed is ignored versus when included in the analysis. As noted in Rybakken et al., 2018, in addition to the circular features in the HD data (same as used in this work), the data gives indications of additional structure that correlates with speed. Perhaps this example could then further illustrate the method with the real data and maybe also shed light on the apparent speed tuning.

4) The authors mention (in line 59) that there exist other methods for structure quantification but do not cite/name them directly or compare them with SI, other than to mention they rely on clustering analysis. Perhaps the authors could include a short discussion either in the introduction or the supplementary about how they differ/complement each other in how they are defined, what they capture, etc. Also, I found the argument that the clustering renders them not useful for continuous variables a bit weak as it is common (and is a natural approximation) to discretize continuous variables or to use some form of clustering into discrete states (e.g. Rubin et al., 2019 looked at the same HD data by first clustering into discrete states). A longer discussion should also help support the argument in lines 70-71, that these “approaches provide poor insight about the local versus global structure of feature representations.”

5) In the head direction example, it seemed strange that the method was used for only one number of neighbors (3), and not varied as in figure 2. It seemed a good opportunity to illustrate the different concepts presented in the text with real data. I suppose they could also compare, e.g. an isomap versus pca embedding of the data, maybe with varying dimensionality (to make support the point made about comparing dimensionality reduction methods).

Overall, I would like to thank the authors for the manuscript. I enjoyed reading it and thinking about the problem.

**Have the authors made all data and (if applicable) computational code underlying the findings in their manuscript fully available?**

Reviewer #1: Yes

Reviewer #2: Yes

PLOS authors have the option to publish the peer review history of their article (what does this mean?). If published, this will include your full peer review and any attached files.

Reviewer #1: No

Reviewer #2: No
---

## [Decision Letter · Decision Letter 1]

20 Nov 2023

Dear Prof. Liset Menendez de la Prida,

Thank you very much for submitting your manuscript "Quantifying the distribution of feature values over data represented in arbitrary dimensional spaces" for consideration at PLOS Computational Biology. As with all papers reviewed by the journal, your manuscript was reviewed by members of the editorial board and by several independent reviewers. The reviewers appreciated the attention to an important topic. Based on the reviews, we are likely to accept this manuscript for publication, providing that you modify the manuscript according to the review recommendations.

Sincerely,

Carina Curto

Guest Editor

PLOS Computational Biology

Thomas Serre

Section Editor

PLOS Computational Biology

Reviewer's Responses to Questions

**Comments to the Authors:**

Reviewer #1: I believe the authors have made adequate changes to the manuscript to rectify the issues that I highlighted in my previous review. However, I find the description of Mapper (requested in the last review, added from line 432 to 447) to be a bit lacking: Mapper is not limited to studying "how the point cloud itself is structured" - it very much is a tool for studying _the relationship between the point cloud AND the filter_ chosen. Moreover, the authors go on to talk about limitations in Mapper software. That may or may not be accurate, but is besides the point. Any overlap between Mapper and the SI metric in the manuscript persists independently on any limitation in available implementations of Mapper. The remaining distinctions (directedness and the availability of a quantifying index) do seem relevant and accurate, and a welcome addition to the manuscript.

Reviewer #2: First, I would like to thank the reviewers for replying to my comments from the first review. I think, generally I am satisfied with the replies and the revision.

Regarding the theoretical basis, I was hoping that it could have been expanded a bit more. I thought the other reviewer’s suggestion about the similarity with Mapper was an excellent way to get started. The authors correctly point out that their proposal is different in that the graph is directed and that they study a measure of the graph. To that, I would say it isn’t clear that the directed aspect of the graph is so critical and if it is, that would be interesting. The examples presented in the figures resulted in graphs that seemed to be mostly symmetric, with the exception of 1(f) and even there for a larger value of k it would probably also start looking more symmetric. Either way, I would still think that starting from Mapper could give the work a nice theoretical basis and then maybe one could justify (perhaps with theoretical or practical arguments) that the graph should be made directed to gain additional information. Again, I would have liked to see something like this and think it would add to the paper but this is not a sticking point for me.

In the introduction, the idea of “global versus local structure” comes up for the first time in the penultimate paragraph and without any explanation as to what is meant by global or local structure. I would suggest adding in a few sentences, especially to support the statement that “existing methods provide poor insights about the local versus global structure”.

The text still has a number of grammatical and spelling errors. I would recommend using a tool or a person who is really picky about those things to go through it.

I enjoyed reading the work again and found it interesting.

**Have the authors made all data and (if applicable) computational code underlying the findings in their manuscript fully available?**

Reviewer #1: Yes

Reviewer #2: Yes

PLOS authors have the option to publish the peer review history of their article (what does this mean?). If published, this will include your full peer review and any attached files.

Reviewer #1: No

Reviewer #2: No

Figure Files:

Data Requirements:

Reproducibility:

References:

---

## [Editor Report · Decision Letter 2]

18 Dec 2023

Dear Dr Menendez de la Prida,

We are pleased to inform you that your manuscript 'Quantifying the distribution of feature values over data represented in arbitrary dimensional spaces' has been provisionally accepted for publication in PLOS Computational Biology.

Best regards,

Carina Curto

Guest Editor

PLOS Computational Biology

Thomas Serre

Section Editor

PLOS Computational Biology

---

## [Editor Report · Acceptance letter]

29 Dec 2023

PCOMPBIOL-D-23-00107R2 

Quantifying the distribution of feature values over data represented in arbitrary dimensional spaces

Dear Dr M de la Prida,

I am pleased to inform you that your manuscript has been formally accepted for publication in PLOS Computational Biology. Your manuscript is now with our production department and you will be notified of the publication date in due course.

With kind regards,

Livia Horvath
